Reference genes selection for transcript normalization in kenaf (Hibiscus cannabinus L.) under salinity and drought stress

Niu Xiaoping 1
Qi Jianmin 1 qijm863@163.com
Chen Meixia 1 2
Zhang Gaoyang 3
Tao Aifen 1
Fang Pingping 1
Xu Jiantang 1
Onyedinma Sandra A. 1
Su Jianguang 4 jgsu@vip.163.com
1 Key Laboratory for Genetics, Breeding and Multiple Utilization of Crops, Fujian Agriculture and Forestry University , Fuzhou , China
2 College of Life Sciences, Ningde Normal University , Ningde , China
3 College of Life Sciences, Shangrao Normal University , Shangrao , China
4 Institute of Bast Fiber Crops, Chinese Academy of Agricultural Sciences , Changsha , China
VanBuren Robert
Electronic publication date: 2015 Nov 26
Publication date: 2015
Volume: 3
Electronic Location ID: e1347
Received 2015 Jul 30; Accepted 2015 Oct 4
Copyright: © 2015 Niu et al.
Copyright year: 2015
Copyright holder: Niu et al.
License: This is an open access article distributed under the terms of the Creative Commons Attribution License, which permits unrestricted use, distribution, reproduction and adaptation in any medium and for any purpose provided that it is properly attributed. For attribution, the original author(s), title, publication source (PeerJ) and either DOI or URL of the article must be cited.
License URL: https://creativecommons.org/licenses/by/4.0/

Keywords: Reference gene, Salinity and drought stress, Gene expression, Kenaf (Hibiscus cannabinus L.)

Funding: 948 project of the Agricultural Department of China 2013-Z70 National bast fiber germplasm resources project of China K47NI201A National bast fiber research system of China CARS-19-E06 This work was funded by the 948 project of the Agricultural Department of China (2013-Z70), National bast fiber germplasm resources project of China (K47NI201A) and National bast fiber research system of China (CARS-19-E06). The funders had no role in study design, data collection and analysis, decision to publish, or preparation of the manuscript.

==============================
Kenaf (Hibiscus cannabinus) is an economic and ecological fiber crop but suffers severe losses in fiber yield and quality under the stressful conditions of excess salinity and drought. To explore the mechanisms by which kenaf responds to excess salinity and drought, gene expression was performed at the transcriptomic level using RNA-seq. Thus, it is crucial to have a suitable set of reference genes to normalize target gene expression in kenaf under different conditions using real-time quantitative reverse transcription-PCR (qRT-PCR). In this study, we selected 10 candidate reference genes from the kenaf transcriptome and assessed their expression stabilities by qRT-PCR in 14 NaCl- and PEG-treated samples using geNorm, NormFinder, and BestKeeper. The results indicated that TUBα and 18S rRNA were the optimum reference genes under conditions of excess salinity and drought in kenaf. Moreover, TUBα and 18S rRNA were used singly or in combination as reference genes to validate the expression levels of WRKY28 and WRKY32 in NaCl- and PEG-treated samples by qRT-PCR. The results further proved the reliability of the two selected reference genes. This work will benefit future studies on gene expression and lead to a better understanding of responses to excess salinity and drought in kenaf.

Introduction

Agricultural productivity worldwide is adversely affected by various environmental stresses, such as water deficiency, excess salinity, extreme temperatures, chemical toxicity, and oxidative stress. These environmental factors can occur at multiple stages of plant development, resulting in reduced productivity and significant crop losses. Worse yet, drought and salinity are becoming particular widespread in many regions, affecting more than 10% of arable land and causing a global decline in the average yields of major crops by more than 50% (Bartels & Sunkar, 2005). Therefore, understanding plant tolerance of drought and salinity is of fundamental importance and has become the focus of intensive research.

Kenaf (Hibiscus cannabinus L.) is an annual herbaceous crop of the Malvaceae family that is mostly grown in tropical regions of Asia and Africa. It has great potential applications in the pulp and paper industry, oil absorption and potting media, board making, filtration media, and animal feed (Ayadi et al., 2011). Kenaf has also been identified as an appealing alternative fiber source for the manufacture of a large range of paper products; pulping kenaf requires less energy and chemical inputs for processing compared with standard wood sources (Bhardwaj, Webber & Sakamoto, 2005; Villar et al., 2009). What is more important is that kenaf has been recognized as a species tolerant of soil salinity and drought (Banuelos, Bryla & Cook, 2002; Curtis & Läuchli, 1985; Francois, Donovan & Maas, 1992). Traditional breeding approaches to improve abiotic stress tolerance have had some successes but are limited by the multigenic nature of the trait and cannot meet commercial demands due to a lack of efficient selection techniques and low levels of genetic variance and fertility (Bartels & Sunkar, 2005). Fortunately, large-scale transcriptomic analyses have recently provided a better understanding of plant stress responses, and several genes have been discovered that respond to drought, salt, or cold stress at the transcriptional level based on gene expression (Chuaqui et al., 2002; Rasmussen et al., 2013; Zhu, 2002). Nevertheless, the analysis of gene expression frequently involves precise and reproducible measurements for given mRNA sequences using Northern blotting or reverse transcription-PCR (RT-PCR). Real-time quantitative RT-PCR (qRT-PCR), which is a high-throughout technique with wide applications to living organisms, has become the preferred method for the validation of microarray results and the quantitation of gene expression (Chuaqui et al., 2002; Vandesompele et al., 2002). To acquire accurate information, real-time RT-PCR results are typically compared to those of an internal reference gene. As internal controls, the expression levels of these reference genes should remain constant across different tissues, experimental treatments, and developmental stages (Bustin, 2002). However, many studies have found that the expression of housekeeping genes varies considerably under different experimental conditions. Thus, it was recommended that the stability of candidate reference genes must be systematically evaluated prior to their use in qRT-PCR normalization (Czechowski et al., 2005; Guenin et al., 2009; Schmidt & Delaney, 2010).

Currently, several statistical algorithms, such as geNorm (Vandesompele et al., 2002), NormFinder (Andersen, Jensen & Orntoft, 2004), BestKeeper (Pfaffl et al., 2004), and the ΔCt method, have been developed to facilitate the identification and assessment of the best set of candidate reference genes; an increasing number of stable reference genes have been screened in many plant species, including EF1 α and FBX in Arabidopsis thaliana (Czechowski et al., 2005; Remans et al., 2008); 18S rRNA, UBQ5 and eEF1 α in Oryza sativa (Narsai et al., 2010); EF1 α in Triticum aestivum (Long et al., 2010), Glycine max (Jian et al., 2008), and Solanum tuberosum (Nicot et al., 2005); FBX and PP2A in Nicotiana benthamiana (Liu et al., 2012); PP2A, ACT4, UBQ14, FBX6, MZA, and PTB in Gossypium hirsutum (Artico et al., 2010); and EF1 α and GADPH in Linum usitatissimum (Huis, Hawkins & Neutelings, 2010). However, no systematic validation of reference genes has been performed in kenaf (H. cannabinus L.), which limits further studies of this species at the functional gene and transcriptomic levels.

In the present study, to identify reliable reference genes that could serve as normalization factors for qRT-PCR data analysis in kenaf under conditions of drought and excess salinity, 10 candidate reference genes (18S rRNA, ACT4, EF1 α, MZA, PP2A, PTB, RAN, TUB α, UBC, and UBQ) were selected and studied their expression stability in a set of 14 kenaf samples exposed to drought and excess salinity using the three statistical algorithms geNorm, NormFinder, and BestKeeper. Furthermore, the transcription factor WRKY28, the homolog of A. thaliana WRKY28 (At4g18170) (Babitha et al., 2013; Chen et al., 2013), was investigated to further validate the accuracy of the novel developed reference gene(s).

Materials and Methods

Plant materials and treatments

Kenaf (H. cannabinus L.) variety Fuhong 992 was used for all experiments. To achieve disease-free materials, seeds were rinsed under running water for 10 min and sterilized with 5% sodium hypochlorite for 10 min, washed three times with sterile water, and then germinated on filter paper that had been saturated with water in complete darkness at 28 °C. After 3 days, seedlings were grown in the greenhouse in 25% Hoagland solution under a 16/8-h light/dark photocycle at 28/26 °C (day/night). The most consistent seedlings at the 3–5 leaf stage were used for excess salinity and drought treatments or for the harvesting of different treated leaf samples. For the excess salinity treatment, seedlings were subjected to 200 mM NaCl and harvested at 2, 4, 6, 8, 12, and 24 h. For the drought treatment, seedlings were treated with 20% (w/v) PEG6000, and samples were collected at several time points (2, 4, 6, 8, 12, and 24 h). Untreated seedlings were harvested as a control at 0 h. Three biological replicates were obtained from each group. All treated-leaf samples were employed in experiments that evaluated the stabilities of candidate reference genes under excess salinity and drought stresses.

Total RNA extraction and cDNA synthesis

All samples were snap-frozen in liquid nitrogen and stored at −80 °C before RNA extraction. The OMEGA isolation kit (R6827-01, USA) was used for RNA extraction, following the elimination of genomic DNA by RNase-free DNase I (TaKaRa, Japan); RNA integrity was then assessed by 2% agarose gel electrophoresis, and RNA sample quality was determined using a NanoDrop 2000 spectrophotometer (NanoDrop, Thermo Scientific). Finally, RNA samples with an A260/A280 ratio of 1.9–2.1, and an A260/A230 ratio greater than 2.0 were used for further analyses. Subsequently, first-strand cDNA was synthesized in a 20-µl reaction using the PrimeScript® RT reagent kit (TaKaRa, Japan) following the manufacturer’s protocol. The quality and integrity of the cDNA were checked by NanoDrop 2000 spectrophotometry and agarose gel electrophoresis, respectively, and stored at −20 °C until use.

Primer design, verification of PCR products, and qRT-PCR

The sequences of 10 candidate reference genes (18S rRNA, ACT4, EF1 α, MZA, PP2A, PTB, RAN, TUB α, UBC and UBQ) from the transcriptomic data (L Zhang, 2015, unpublished data) of kenaf were subjected to a BLAST search using A. thaliana sequences in GenBank. The probe sequences from A. thaliana were used to search the transcriptome of kenaf and the kenaf expressed sequence tags database (http://www.ncbi.nlm.nih.gov/nucest/?term=kenaf). Target sequences were identified by querying homologous kenaf sequences together with A. thaliana complete CDS and probe sequences. According to these potential reference gene sequences, primers were designed using Primer 3 (http://bioinfo.ut.ee/primer3-0.4.0/primer3/) based on the following criteria: GC content 45–80%, melting temperatures 58–62 °C, primer lengths 18–24 bp, and amplicon lengths 100–250 bp (see Table 1 for detailed primer information). To check the specificity of the amplicon, all primer pairs were initially tested via standard RT-PCR using the Premix Ex Taq™(TaKaRa, Japan); the amplification products of each gene were verified by 2% agarose gel electrophoresis. Real-time amplification reactions were performed with the Applied Biosystems 7500 Real-Time PCR System using SYBR® Premix Ex Taq. Reactions were prepared in a 20-µl volume containing 2 µl cDNA template, 0.4 µl each amplification primer, 0.4 µl ROX Reference Dye II, 10 µl 2 × SYBR Premix Ex Taq, and 6.8 µl dH2O. Amplifications were performed with an initial step of 95 °C for 30 s, followed by 40 cycles of denaturation at 95 °C for 5 s and primer annealing at 60 °C for 34 s. The melting curve ranged from 60 °C to 95 °C, and the temperature was increased in increments of 0.2 °C per 10 s for all PCR products. ABI Prism Dissociation Curve Analysis Software was used to confirm the occurrence of specific amplification peaks. All reactions were carried out in triplicate with template-free negative controls being performed in parallel.

Table 1 Primer sequences and amplicon characteristics of the 10 candidate internal control genes evaluated in this study.

Gene	A. thaliana ortholog locus	Gene description	Primer sequence F/R (5′–3′)	Product size (bp)	Efficiency (%)	Mean	CV (%)	
						R 2	Ct	SD		
18S rRNA	At3g41768	18S ribosomal RNA	CTACGTCCCTGCCCTTTGTA	175	104.1	0.9949	17.89	1.546	8.64	
			GGTTCACCTACGGAAACCTTG							
ACT4	At5g09810	Actin 4	TTGCAGACCGTATGAGCAAG	166	105.4	0.9967	22.17	1.240	5.59	
			ATCCTCCGATCCAGACACTG							
MZA	At5g46630	Clathrin adaptor complexes	CCGTCAGACAGATTGGAGGT	154	106.3	0.9949	34.28	0.7416	2.16	
		medium subunit family protein	AAAGCAACAGCCTCAACGAC							
EF1α	At5g60390	Elongation factor 1-alpha	TCCCCATCTCTGGTTTTGAG	130	113.8	0.9960	22.46	1.542	6.87	
			CTTGGGCTCATTGATCTGGT							
RAN	At5g59840	Ras-related small	GCCATGCCGATAAGAACATT	167	97.13	0.9997	32.18	1.151	3.58	
		GTP-binding protein	GTGAAGGCAGTCTCCCACAT							
TUBα	At4g14960	Alpha-tubulin	AATGCTTGCTGGGAGCTTTA	213	105.1	0.9992	30.24	0.6500	2.15	
			GTGGAATAACTGGCGGTACG							
PP2A	At1g59830	Catalytic subunit of protein	GATCCTTGTGGAGGAGTGGA	201	108.9	0.9985	28.38	1.149	4.05	
		phosphatase 2A	GCGAAACAGTTCGACGAGAT							
UBC	At3g52560	Ubiquitin-conjugating	CTGCCATCTCCTTTTTCAGC	150	118.6	0.9981	23.72	1.169	4.93	
		enzyme like protein	CGAGTGTCCGTTTTCATTCA							
PTB	At3g01150	Polypyrimidine tract-binding	GGTTACCATTGAGGGTGTGG	158	109.4	0.9993	28.14	1.321	4.69	
		protein homolog	GTGCACAAAACCAAATGCAG							
UBQ	At5g20620	Ubiquitin	TCTTTGCAGGGAAGCAACTT	219	102.5	0.9993	31.15	1.944	6.24	
			CTGCATAGCAGCAAGCTCAC							

Statistical analyses

To select a suitable reference gene, geNorm, NormFinder, and BestKeeper were used to analyze the stability of each candidate reference gene. All analyses using these packages were performed according to the manufacturer’s instructions. The raw Ct values from qRT-PCR were transformed into the required data input format. The maximum expression level (the minimum Ct value) of each gene was used as a control and was set to a value of 1. Then relative expression levels were calculated from Ct values using the formula: 2−ΔCt, where ΔCt = each sample Ct value—minimum Ct value. The geNorm and NormFinder algorithms further calculated the expression stability value (M) for each gene with the obtained data, whereas the BestKeeper analyses were based on untransformed Ct values. Standard curves were generated using Microsoft Excel 2003 by plotting cycles at threshold fluorescence (Ct) against the logarithmic values of standard RNA amounts. Quantities of standard RNA were prepared by diluting cDNA (1, 10−1, 10−2, 10−3, 10−4, and 10−5; each gene in triplicate). Only Ct values less than 40 were used to calculate correlation coefficients (R2 values) and amplification efficiencies (E) from the slope generated in Microsoft Excel 2003, according to the equation E = [10−(1/slope) − 1] × 100%. All PCR assays showed efficiency values between 97% and 118% (Table 1). All other multiple comparisons were performed with the statistical analysis software SPSS 22.0 (SPSS Inc., USA). To validate the reference gene(s) selected in the current study, the relative expression level of WRKY28 was normalized using the 2−ΔΔCt method after collecting the mean Ct value of each biological replicate from the samples treated under conditions of salinity or drought for 0, 4, 6, 8, 12, and 24 h. Finally, the relative increases in expression level of WRKY28 were used to calculate the differences in the normalization of each reference gene.

Results

Selection of candidate reference genes, primer specificity, and amplification efficiency

A total of 10 candidate reference genes, including four commonly used housekeeping genes, 18S rRNA, ACT4, TUB α, and UBQ, and six novel candidate reference genes, EF1 α, MZA, PP2A, PTB, RAN, and UBC, were identified in this study (Table 1). The four novel candidates were validated in A. thaliana, O. sativa, or G. hirsutum for expression stabilities under different abiotic stresses. RAN and UBC have been evaluated as the optimal reference gene in Cucumis melo (Kong et al., 2014) and Platycladus orientalis (Chang et al., 2012), respectively. Additionally, the specificity of the designed primers was verified by a single band with the expected size after agarose gel electrophoresis (Fig. S1). Specificity was further confirmed by a single peak in the melting curve analysis, which was done prior to performing qRT-PCR (Fig. S2). A standard curve was generated using a 10-fold dilution of cDNA in the qRT-PCR assay to determine the amplification efficiency for each primer pair. Both E and R2 were calculated using the slope of the standard curve. Results indicated that the average amplification efficiency values of all primers ranged from 97.13% to 118.60% (Table 1).

As shown in Fig. 1 and Table 1, for all tested samples, the mean Ct values of 10 candidate reference genes had a wide range (17.89–34.28), and the standard deviation (SD) varied from 0.65 to 1.94, and the coefficient of variation (CV) ranged from 2.15% to 8.64%. Comparing to Ct values of all the candidates, 18S rRNA had a highest expression level (mean Ct ± SD = 17.89 ± 1.55), following ACT4 (mean Ct ± SD = 22.17 ± 1.24) and EF1 α (mean Ct ± SD = 22.46 ± 1.54), whereas MZA (mean Ct ± SD = 34.28 ± 0.74) accumulated less than all the others. Additionally, PP2A, PTB, RAN, TUB α, UBQ and UBC showed a narrow range of mean Ct values, indicating that these genes were stably expressed. However, it is insufficient for evaluating gene expression stability just by the comparison of raw Ct values. To obtain accurate gene expression data, other approach should be combined with to validate a set of appropriate reference genes under a given conditions.

Figure 1 Expression levels of 10 candidate reference genes across all experimental samples.

Each box indicates 25/75 percentiles. Whisker caps represent 10/90 percentiles. The median is depicted by the line across the box, and all outliers are indicated by dots.

Stability of reference genes analysis by geNorm

The geNorm-based analysis was conducted to determine which reference gene(s) would be optimal in each tested sample set. As shown in Fig. 2, the candidate genes were ranked accordingly to their M values. Since a lower M values indicates a greater stability of the reference gene, TUB α and 18S rRNA were ranked as the most stable, with M values of 0.28 and 0.65, respectively, whereas UBC was the least stably expressed under conditions of drought and excess salinity (Fig. 2). The results were consistent with the pattern observed in Tables 2 and 3. When the combination of drought and salinity was considered, the same results (TUB α and 18S rRNA) were acquired for normalization (M = 0.51) (Table 4 and Fig. 2C). The geNorm algorithm can also be used to identify the optimal number of reference genes by calculating the pairwise variation (V) between normalization factors (NFn). It is proposed that an additional reference gene makes no sense to the normalization when Vn/n+1 is less than 0.15. In this study, the data showed that a V2/3 of 0.13 was less than 0.15, which indicated that the combination of TUB α and 18S rRNA was sufficient for the normalization of gene expression under drought stress (Fig. 2D). For salinity stress in kenaf, a V3/4 of 0.15 was less than a V2/3 of 0.19, which suggested that three reference genes, TUB α, 18S rRNA, and RAN were the best options for accurate normalization under salinity stress (Fig. 2D). For all drought and excess salinity samples, the same effects were observed. These results revealed that TUB α together with 18S rRNA (V2/3 = 0.14) could provide a reliable reference for the normalization of gene expression.

Figure 2 geNorm ranking of candidate reference genes and pairwise variation (V) to determine the optimal number of reference genes.

(A) Expression stability of 10 candidate reference genes after NaCl treatment. (B) Expression stability of 10 candidate reference genes after PEG treatment. (C) Expression stability of 10 candidate reference genes after NaCl and PEG treatments. (D) The pairwise variation (Vn/Vn + 1) was calculated between normalization factors NFn and NFn+1 by geNorm to determine the optimal number of reference genes for accurate normalization.

Table 2 Expression stability of H. cannabinus candidate reference genes under salinity stress.

Rank	geNorm	NormFinder	BestKeeper	
	Gene	Stability	Gene	Stability	Gene	CV ± SD	r	p-value	
1	TUBα	0.65	PP2A	0.11	TUBα	1.50 ± 0.46	−0.186	0.692	
2	18S rRNA	0.65	ACT4	0.11	18S rRNA	1.96 ± 0.67	0.911	0.004	
3	RAN	0.67	18S rRNA	0.40	RAN	1.97 ± 0.63	0.307	0.504	
4	UBQ	0.70	PTB	0.51	PP2A	3.38 ± 0.94	0.949	0.001	
5	ACT4	0.87	UBQ	0.56	MZA	3.99 ± 0.67	0.014	0.977	
6	PP2A	0.94	RAN	0.83	UBQ	3.99 ± 1.28	0.691	0.085	
7	MZA	1.10	TUBα	0.93	PTB	4.53 ± 1.25	0.958	0.001	
8	PTB	1.19	MZA	1.14	UBC	5.15 ± 1.20	0.952	0.001	
9	EF1α	1.41	EF1α	1.35	EF1α	6.30 ± 1.36	0.943	0.001	
10	UBC	1.68	UBC	1.91	ACT4	6.33 ± 1.38	0.959	0.001	

Table 3 Expression stability of H. cannabinus candidate reference genes under drought stress.

Rank	geNorm	NormFinder	BestKeeper	
	Gene	Stability	Gene	Stability	Gene	CV ± SD	r	p-value	
1	18S rRNA	0.28	18S rRNA	0.12	MZA	0.72 ± 0.25	0.246	0.593	
2	TUBα	0.28	TUBα	0.13	TUBα	1.54 ± 0.46	0.837	0.019	
3	ACT4	0.38	ACT4	0.18	PP2A	2.21 ± 0.64	0.424	0.342	
4	MZA	0.42	UBQ	0.31	ACT4	2.32 ± 0.52	0.819	0.024	
5	UBQ	0.51	MZA	0.34	PTB	2.40 ± 0.69	0.601	0.154	
6	PP2A	0.62	PP2A	0.65	UBC	2.65 ± 0.64	0.766	0.044	
7	RAN	0.71	PTB	0.68	EF1α	3.15 ± 0.74	0.789	0.035	
8	PTB	0.80	RAN	0.76	RAN	3.50 ± 1.13	0.871	0.011	
9	EF1α	0.99	EF1α	1.08	UBQ	3.72 ± 1.12	0.640	0.122	
10	UBC	1.13	UBC	1.09	18S rRNA	4.47 ± 0.85	0.940	0.002	

Stability of reference genes analysis by NormFinder

The NormFinder program analyzes candidate reference genes according to inter- and intragroup variations in expression. As in the geNorm method, the gene with the lowest M value has the most stable expression. As shown in Tables 2–4, the NormFinder analysis also identified that 18S rRNA and TUB α were the most stably expressed genes with values of 0.12 and 0.13 under drought stress, respectively, with slight differences in the ranking order (Table 3). For the salinity samples, PP2A and ACT4 were the top two reference genes followed by 18S rRNA, identified by NormFinder (Table 2). For the drought and excess salinity samples, ACT4 and 18S rRNA were the top two genes calculated by NormFinder (Table 4). Compared with the results of geNorm and NormFinder analyses, 18S rRNA was the best reference gene under salinity stress. Overall, the results obtained from NormFinder analysis were consistent with those from geNorm analysis.

Table 4 Expression stability of H. cannabinus candidate reference genes under salinity and drought stress.

Rank	geNorm	NormFinder	BestKeeper	
	Gene	Stability	Gene	Stability	Gene	CV ± SD	r	p-value	
1	TUBα	0.51	ACT4	0.20	MZA	1.31 ± 0.45	0.117	0.689	
2	18S rRNA	0.51	18S rRNA	0.34	TUBα	1.89 ± 0.57	−0.100	0.732	
3	UBQ	0.61	UBQ	0.41	RAN	2.66 ± 0.86	0.561	0.037	
4	RAN	0.74	PP2A	0.42	PP2A	3.24 ± 0.92	0.832	0.001	
5	ACT4	0.82	PTB	0.58	UBC	3.88 ± 0.92	0.903	0.001	
6	PP2A	0.89	TUBα	0.64	PTB	3.89 ± 1.09	0.883	0.001	
7	MZA	0.97	RAN	0.76	ACT4	4.42 ± 0.98	0.910	0.001	
8	PTB	1.06	MZA	0.90	EF1a	4.89 ± 1.10	0.915	0.001	
9	EF1α	1.41	EF1α	1.30	UBQ	5.44 ± 1.70	0.243	0.401	
10	UBC	1.61	UBC	1.62	18S rRNA	6.84 ± 1.22	0.903	0.001	

Stability of reference genes analysis by BestKeeper

The BestKeeper program utilizes the unconverted Ct values to accomplish parametric tests on normally distributed expression levels. It evaluates the geometric mean of Ct values, determines the coefficient of variance (CV), and calculates a Pearson’s coefficient of correlation (r) of each gene. The standard deviation (SD) of the average Ct values are performed to generate a weighted index of most suitable normalizing genes across selected biological samples and exclude genes that are not stably expressed. The most stable genes are identified as those that exhibit the lowest CV and SD (CV ± SD), in relation to r and p values. Genes with SD greater than 1 are considered unacceptable and should be excluded. In this study, 18S rRNA had CV ± SD values of 1.96 ± 0.67, showing remarkably stable expression (r = 0.911; p = 0.004). TUB α displayed the lowest CV ± SD values of 1.50 ± 0.46, but with a lower r value (r = − 0.186), and higher p-value (p = 0.692), it is not possible to conclude on stable gene. UBQ, PTB, UBC, EF1 α and ACT4 had an SD > 1.0, thus excluding them as suitable reference genes for NaCl-treated samples (Table 2). These results are consistent with those acquired from the geNorm method. For the PEG-treated samples, MZA displayed the lowest CV ± SD = 0.72 ± 0.25, but with r = 0.246, p = 0.593, it does not act as a suitable reference gene. TUB α (CV ± SD = 1.54 ± 0.46, r = 0.837, and p = 0.019) and 18S rRNA (CV ± SD = 4.47 ± 0.85, r = 0.940, and p = 0.002) showed the potential for being normalization factors. Whereas UBQ and RAN had an SD value greater than 1.0, indicating that the two reference genes were not suitable for normalization (Table 3). When the two datasets from conditions of excess salinity and drought were analyzed together, although MZA and TUB α had a lower CV ± SD, a lower r value and higher p value was observed. It is notable that 18S rRNA with SD >1.0 was identified as the least stable by BestKeeper compared to the other candidate reference genes (Tables 3 and 4).

Reference gene validation

To further validate the reliabilities of the selected reference genes in the current study, the relative expression level of two transcription factors, HcWRKY28 and HcWRKY32, were investigated in two kenaf samples under conditions of drought and excess salinity, using one or two of the most stable reference genes, as well as the least stable gene for normalization, which had been determined by geNorm, NormFinder, and BestKeeper as described above (Tables 2–4 and Fig. 2). In A. thaliana, AtWRKY28 (At4g18170) plays a crucial role in protecting plants against pathogen infections and oxidative stress. In our previous study, we found that kenaf WRKY28, a homolog of AtWRKY28, was differentially expressed under abiotic stress conditions according to the gene differential expression profiling analysis based on RNA-seq. We also found that WRKY32 was actively upregulated in kenaf in response to drought and salinity stresses (X Niu, 2015, unpublished data). Therefore, the two genes were selected to further validate the usefulness of the selected reference genes by qRT-PCR. In the current study, we selected the two reference genes with lowest M values, 18S rRNA and TUB α and their combination (18S rRNA + TUB α), and the gene (UBC) with highest M value, as the normalization factors to determine the expression patterns. As shown in Fig. 3, the expression abundance of WRKY28 increased significantly after 6 h of salinity treatment, peaked at 8 h, and thereafter decreased (P < 0.01) (Fig. 3A), while WRKY32 peaked at 12 h (P <0.01) (Fig. 3C) in the NaCl-treated samples. For PEG-treated samples, qRT-PCR analysis showed that the expression level of WRKY28 decreased from 0 h to 6 h, increased at 8 h, peaked at 12 h, and then decreased (P < 0.01) (Fig. 3B). In contrast, the expression level of WRKY32 increased at 12 h, and peaked at 24 h with approximately 7.5-fold accumulation (P < 0.01) (Fig. 3D). Salinity and drought stresses significantly increased the expression of WRKY28 by 4.5- and 3.5-fold at 8 h and 12 h, WRKY32 by 6.5- and 7.5-fold at 12 h and 24 h, compared with that at 0 h, respectively (P < 0.01). These results are in accordance with the patterns of WRKY28 and WRKY32 in previously analysis. Moreover, the expression patterns of WRKY28 and WRKY32 showed similar trends across different experimental sets to those of 18S rRNA and TUB α as reference genes either singly or in combination (Fig. 3), although some slight differences exist. However, when the expression of WRKY28 and WRKY32 was normalized using the reference gene with the highest M value (UBC) as an internal control, the expression pattern of WRKY28 and WRKY32 showed a significantly different changes, and the expression level was 5-fold lower than that of the genes with the lowest M values (P < 0.01) (Fig. 3). These results indicate that 18S rRNA and TUB α are suitable reference genes for gene expression normalization under conditions of drought and excess salinity in kenaf.

Figure 3 Relative increase in expression of WRKY28 and WRKY32 using the selected reference genes.

Relative expression of WRKY28 (A, B) and WRKY32 (C, D) was normalized using the most stable reference genes: single TUBα, single 18S rRNA, or TUBα + 18S rRNA in sample sets across NaCl-treated samples (A and C) and PEG- treated samples (B and D). cDNA samples were taken from the same set used for gene expression stability analysis.

Discussion

Water deficiency and high salinity are responsible for the large margin existing between potential and real harvest yields in several crops worldwide (Mittler, 2006; Wang, Vinocur & Altman, 2003). To adapt to adverse conditions, plants have evolved complicated mechanisms at multiple levels to increase stress tolerance. At the molecular level, some stress-response and -tolerant genes contribute to the plants’ ability to cope with unfavorable environmental conditions (Rasmussen et al., 2013; Zhu, 2002). Many studies on plant defense and stress mechanisms are increasingly based on gene expression analyses (Chuaqui et al., 2002). Real-time qRT-PCR has been widely used as an accurate and reliable method for the detection of gene expression and plays a considerable role in the study of molecular mechanisms of plant stress responses. To avoid bias caused by RNA integrity, cDNA synthesis, and tissue or cell activities, qRT-PCR uses internal reference genes, which should not fluctuate under different experimental treatments. However, many studies have found that the expression of reference genes vary considerably across different experimental conditions, and thus, it is proposed that the stability of candidate reference genes should be systematically evaluated prior to their use in qRT-PCR normalization (Czechowski et al., 2005; Guenin et al., 2009; Schmidt & Delaney, 2010). Kenaf is well known as a cellulosic source with both economic and ecological advantages. Recently, it has been used as an alternative raw material to wood in the pulp and paper industry to limit forest destruction and as nonwoven mats in the automotive and textile industries (Nishino et al., 2003; Villar et al., 2009). Therefore, breeding kenaf cultivars with tolerance to drought and salinity is of fundamental importance, and understanding plant response mechanisms to drought and salinity would greatly contribute to increases crop yield and efficient land usage (Mittler, 2006; Suzuki et al., 2014; Wang, Vinocur & Altman, 2003). However, the limited available of suitable reference genes, which should be selected under systematic and specific experimental conditions, has allowed few applications of qRT-PCR for kenaf functional gene characterization.

In this study, we used the three different programs (geNorm, NormFinder and BestKeeper) to systematically evaluate the expression stability of candidate reference genes in kenaf under conditions of drought and excess salinity. Differences were observed among these three algorithms after comparisons of the ranking of the candidate reference genes (Tables 2–4 and Fig. 2). This is an expected event due to the different statistical algorithms used to calculate stability associated with each method (Liu et al., 2012; Nicot et al., 2005). In addition, variations in single or combined stresses are indicative of the expression levels of reference genes and their responses to different stresses. Recent studies have revealed that the molecular and metabolic responses of plants to simultaneous stresses are unique and cannot be directly extrapolated from the responses of each of the stresses applied individually (Mittler, 2006; Wang, Vinocur & Altman, 2003). After the analysis of data from salinity-treated samples, TUB α, 18S rRNA, and RAN were ranked as the most stable reference genes by geNorm and BestKeeper (Table 2 and Fig. 2A). For drought-treated samples, the top two reference genes (TUB α and 18S rRNA) identified by geNorm were the same as those determined by NormFinder, but not those assigned by BestKeeper (Table 3 and Fig. 2B). In term of NaCl- and PEG-treated samples, TUB α and 18S rRNA were ranked as the top two reference genes by geNorm (Table 4 and Fig. 2C), ACT4 and 18S rRNA by NormFinder, and MZA and TUB α by BestKeeper (Table 4).

Taken together, the results obtained from the three algorithms identified TUB α and 18S rRNA as the best normalization factors in kenaf under conditions of drought and excess salinity. In this study, TUB α was identified as one of the top two reference genes, which was consistent with results in the developmental stages of G. max (Jian et al., 2008), biotic stress samples of C. melo (Kong et al., 2014), and PEG-treated samples of P. orientalis (Chang et al., 2012); however, this gene performed poorly in studies of L. usitatissimum (Huis, Hawkins & Neutelings, 2010) and Phyllostachys edulis (Fan et al., 2013), suggesting that the expression levels of reference genes are variable among different species. 18S rRNA, one of the most stable reference genes in the present study, was also the best reference gene during late blight exposure of S. tuberosum (Nicot et al., 2005), across leaf and root tissues of Cichorium intybus (Maroufi, Van Bockstaele & De Loose, 2010), and in barley yellow dwarf virus-infected cereals (Jarosova & Kundu, 2010), but it was the worst performer in P. orientalis (Chang et al., 2012), P. edulis (Fan et al., 2013), and viral infection of N. benthamiana (Liu et al., 2012). ACT4 was ranked as the third most stable reference gene analyzed by geNorm and NormFinder in samples subjected to drought stress (Table 3); this gene was also optimal for the different floral organs of G. hirsutum (Artico et al., 2010) and in diploid and tetraploid Chrysanthemum nankingense (Wang et al., 2015). In addition, MZA showed remarkably consistent expression during fruit development of G. hirsutum (Artico et al., 2010) and P. edulis (Fan et al., 2013) (namely CAC), and in the development of Solanum lycopersicon (Exposito-Rodriguez et al., 2008) (namely CAC); these results were consistent with those calculated by BestKeeper in samples subjected to drought, as well as in the NaCl- and PEG-treated samples in our present study (Tables 3 and 4). The data also demonstrate that the expression levels of reference genes PP2A, RAN, UBQ, and PTB were considerable variable in this study (Tables 2–4 and Fig. 2). PP2A was reported to be the most stable reference gene in the viral infection of N. benthamiana (Liu et al., 2012), different organs of G. hirsutum (Artico et al., 2010), and in PEG- and heat-treated samples of Caragana intermedia (Zhu et al., 2013), but was unsatisfactory as a reference gene in C. melo (Kong et al., 2014). RAN was also the optimal performer in samples of C. melo treated with growth regulators (Kong et al., 2014) but was variable under conditions of excess salinity and drought in this study. UBQ, one of the most commonly used reference genes in qRT-PCR assays, showed variable expression in our study but was previously reported to be expressed stably in different organs of G. hirsutum (Artico et al., 2010) and in NaCl- and ABA-treated samples of P. orientalis (Chang et al., 2012). A novel reference gene identified in A. thaliana, PTB (Remans et al., 2008), has been reported as one of the most stably expressed gene in fruit development of G. hirsutum (Artico et al., 2010), but it was not suitable as a normalization factor in kenaf under conditions of excess salinity and drought according to the three algorithms. Previously, EF1 α was the most highly recommended reference gene during biotic and abiotic stresses in both S. tuberosum (Nicot et al., 2005) and G. max (Saraiva et al., 2014). In this study, however, EF1 α was ranked at the bottom by the three programs used. Similarly, the commonly used reference gene UBC was the least stable gene in analyses from geNorm and NormFinder in all samples. Given these observations, we suggest that both EF1 α and UBC should be carefully used as reference genes in kenaf under conditions of excess salinity and drought.

Finally, to validate the suitability of the reference genes revealed in this study, we analyzed the transcription profile of two WRKY genes, HcWRKY28 and HcWRKY32, in kenaf. HcWRKY28 is a homolog of AtWRKY28, which has been demonstrated as a key plant regulator against pathogen infections and oxidative stress in A. thaliana (Babitha et al., 2013; Chen et al., 2013). In this study, the expression of HcWRKY28 and HcWRKY32 was normalized using the most stable reference genes (TUB α, 18S rRNA, and a combination of both) and a less stable gene (UBC) as internal controls in kenaf under conditions of excess salinity and drought. The results showed that WRKY28 and WRKY32 expression were actively induced by NaCl and PEG treatments (Fig. 3) and were significantly increased after 6 h of NaCl treatment, peaking at 8 h and 12 h (Figs. 3A and 3C) (P < 0.01), respectively; for drought samples, the expression level of WRKY28 decreased from 0 h to 6 h, increased at 8 h, and peaked at 12 h, WRKY32 increased at 12 h, and peaked at 24 h (Figs. 3B and 3D) (P < 0.01). Salinity and drought stress significantly increased the expression of WRKY28 by 4.5- and 3.5-fold at 8 h and 12 h, respectively (P < 0.01), compared to the control group, while WRKY32 by 6.5- and 7.5-fold at 12 h and 24 h, respectively (P < 0.01). Moreover, the expression patterns of WRKY28 and WRKY32 showed similar trends under conditions of excess salinity and/or drought to those of TUB α and 18S rRNA as reference genes, either singly or in combination (Fig. 3), with some slight differences. However, the data showed that the use of the most variable reference gene UBC resulted in significantly expression changes in WRKY28 and WRKY32 compared to the normalized expression data obtained using TUB α, 18S rRNA, or a combination of both. These results indicate that the inappropriate utilization of reference genes without validation may generate bias in the analysis and lead to misinterpretation of qRT-PCR data.

Conclusion

To screen appropriate reference genes to normalize expression of target genes in kenaf under conditions of excess salinity and drought, 10 candidate reference genes were validated across 14 NaCl- and PEG-treated kenaf samples using three commonly used statistical algorithms: geNorm, NormFinder, and BestKeeper. The results revealed that TUB α and 18S rRNA were identified as suitable reference genes for gene expression normalization in kenaf under conditions of excess salinity and/or drought. Furthermore, the expression analyses of WRKY28 and WRKY32 further validated that the combination of TUB α and 18S rRNA as normalization factors was optimal in the present study. This study can benefit future research on the expression of genes in response to excess salinity and drought in kenaf.

Supplemental Information

Supplemental Information 1 PCR products of 10 reference genes checked on a 2.0% agarose gel

Click here for additional data file.

Supplemental Information 2 Melting curves of the 10 candidate reference genes tested in this study

Click here for additional data file.

Figure S1 Raw data for Fig. 1

Click here for additional data file.

Figure S2 Raw data for Fig. 2

Click here for additional data file.

Figure S3 Raw data for Fig. 3

Click here for additional data file.

Table S2 Raw data for Table 2

Click here for additional data file.

Table S3 Raw data for Table 3

Click here for additional data file.

Table S4 Raw data for Table 4

Click here for additional data file.

Additional Information and Declarations

Competing Interests

Author Contributions

The authors declare there are no competing interests.

Xiaoping Niu conceived and designed the experiments, performed the experiments, analyzed the data, contributed reagents/materials/analysis tools, wrote the paper, prepared figures and/or tables, reviewed drafts of the paper.

Jianmin Qi conceived and designed the experiments, wrote the paper, reviewed drafts of the paper.

Meixia Chen conceived and designed the experiments, performed the experiments, prepared figures and/or tables.

Gaoyang Zhang performed the experiments.

Aifen Tao analyzed the data.

Pingping Fang contributed reagents/materials/analysis tools.

Jiantang Xu analyzed the data, contributed reagents/materials/analysis tools.

Sandra A. Onyedinma contributed reagents/materials/analysis tools, reviewed drafts of the paper.

Jianguang Su wrote the paper.

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
