# Peer review of "Reference genes selection for transcript normalization in kenaf (Hibiscus cannabinus L.) under salinity and drought stress"

_PeerJ, doi:10.7717/peerj.1347_

## Round 0.1 · original submission · Major Revisions

Your manuscript entitled “Reference genes selection for transcript normalization in Kenaf (Hibiscus cannabinus L.) under salinity and drought stress” has been seen by two qualified reviewers. As you will see in the comments, both reviewers felt this manuscript was largely technically sound and well suited for publication to PeerJ. However both reviewers have raised some concerns that must be addressed before a decision can be reached. In particular, the authors need to address the technical issues raised by reviewer 1 (including proving evidence of a negative control to test for contaminated gDNA) and address the status of the transcriptome data as requested by reviewer 2.

Reviewer 1 ·

Basic reporting

The quality of the written English is very high. The Introduction and Discussion adequately describe the purpose and importance of the research.

This is a technical paper that will be of use to a limited audience, but the use of proper controls is important such that the information is useful

The article further serves to evaluate, incidentally, the software programs used for analysis.

Experimental design

The experimental design was appropriate, however some clarification is needed and some additional experimentation is needed.

1) Line 93-94, Please clarify what exactly are the "samples" obtained? Are they whole plant, leaves, roots, etc?

2) After DNAse treatment I do not see strong evidence (sufficient evidence) that contaminating gDNA was removed. The methods described (agarose gel and spectrophotometer) do not address contamination. The authors need to at least conduct a negative RT reaction whereby DNAse digested RNA is subjected to a mock cDNA synthesis without Reverse Transcriptase. The resulting reaction is then used for PCR as usual. In this case the Ct value for the negative reactions could be compared to control reactions as evidence for a "successful digestion".

3) Lines 132-136 need clarification. It is unclear how the authors conducted calculations. For example the phrase "maximum expression level" is ambiguous. Does this refer to a gene with lowest Ct, or was a delta Ct used? Further, use of the phrase "corresponding Ct" and "minimum Ct" are unclear as to what Ct value is being used. This whole description needs to be clarified. Perhaps the authors could generate a formula or example to illustrate what they are doing here.

Validity of the findings

The data is generally sound and informative, but some findings should be clarified.

1) In the section "Reference gene expression profiles"

There seems to be a high degree of relevance given to actual, raw Ct values, when in fact these are generally useless. They are completely subject to the individual sample RNA isolation and the efficiency of each cDNA synthesis reaction. Thus, statements like "variations in reference gene mRNA levels were indicated by the range of Ct values..." (Line 169-170) is completely false. Also, lines 172-173, "18S rRNA was the most abundantly expressed gene of all samples with a Ct value less than 18..." Is completely inaccurate. Ct values are only informative when compared to another Ct value. The authors use Ct values to calculate %CV, which does not indicate variability of the gene, but just the efficiency of the mRNA isolation/cDNA reaction.
I cannot think of any useful reason to discuss Ct values, however if the authors want to get some useful information from Ct values then this entire section needs to be rewritten to acknowledge (and not overstate) the meaning of a Ct value.

2) Without more information on the program "BestKeeper" this program does not seem legitimate to use for this purpose. If it only looks at only looks at "stability" of Ct values, then as described above, that is useless.

3) In the section "Reference gene validation"

The data should have at least one more, possible 2 test genes showing the same results. One gene does not show a reliable trend.

Line 245 to 246, You should refer to your reference genes in terms of them having low or high M values - that is what is important. Based on your previous writing, using the phrase "stable" and "variable" implies you selected them only because of the raw Ct value.

Additional comments

These papers showing good reference genes are useful for those doing qPCR with your organism.

Reviewer 2 ·

Basic reporting

Dear Editor,

Thanks for inviting me to peer review manuscript entitled “Reference genes selection for transcript normalization in Kenaf (Hibiscus cannabinus L.) under salinity and drought stress” submitted by Niu et al. for possible publication in PeerJ.

The manuscript present the set of the reference genes, which can be used for the qRT-PCR normalization. Although the manuscript is written well and the data has been discussed with the previous findings. I have few questions to ask the authors before coming to the judgement of the paper.

Authors describe that they have collected the sequences from the transcriptome under the salinity and drought conditions in the paper at line 107, 108. So i wish to ask that why authors have not reported the entire transcriptomics gene profiling and pathways discussion and also the details of the transcriptomic assembly. Is this a part of the transcriptomics paper?, which have been submitted elsewhere. Did author have sequenced the transcriptome or they took from the previously published studies. There is no reference at line 107,108.
If this is a sub-part of the transcriptome paper then i encourage authors to detail the transcriptome dataset here and also the NCBI archive SRA number.

I recommend the publication of the manuscript but wondering that if this is a part of the transcriptomics paper then the gene profiling in the transcriptomics paper might be similar to the same here as there they might have used the same reference genes for the evaluation.

Best wishes,
Reviewer.

Experimental design

Experimental Design is robust and needs no revisions. However, the presentation of the results needs a substantial amount of clarity.

Validity of the findings

The paper lacks the clarity, which i have raised in my comments.

Additional comments

Dear Editor,

Thanks for inviting me to peer review manuscript entitled “Reference genes selection for transcript normalization in Kenaf (Hibiscus cannabinus L.) under salinity and drought stress” submitted by Niu et al. for possible publication in PeerJ.

The manuscript present the set of the reference genes, which can be used for the qRT-PCR normalization. Although the manuscript is written well and the data has been discussed with the previous findings. I have few questions to ask the authors before coming to the judgement of the paper.

Authors describe that they have collected the sequences from the transcriptome under the salinity and drought conditions in the paper at line 107, 108. So i wish to ask that why authors have not reported the entire transcriptomics gene profiling and pathways discussion and also the details of the transcriptomic assembly. Is this a part of the transcriptomics paper?, which have been submitted elsewhere. Did author have sequenced the transcriptome or they took from the previously published studies. There is no reference at line 107,108.
If this is a sub-part of the transcriptome paper then i encourage authors to detail the transcriptome dataset here and also the NCBI archive SRA number.

I recommend the publication of the manuscript but wondering that if this is a part of the transcriptomics paper then the gene profiling in the transcriptomics paper might be similar to the same here as there they might have used the same reference genes for the evaluation.

Best wishes,
Reviewer.

---

## Round 0.2 · accepted · Accept

The reviewers have re-evaluated your revised manuscript. The revisions have addressed the previous concerns and I feel it is now suitable for publication in PeerJ.

Reviewer 1 ·

Basic reporting

After revision the author's have addressed the concerns by the reviewers.

Experimental design

After revision the author's have addressed the concerns by the reviewers.

Validity of the findings

After revision the author's have addressed the concerns by the reviewers.